# Exploring Multi-Channel GPS Receivers for Detecting Spoofing Attacks on UAVs Using Machine Learning

**DOI:** 10.3390/s25134045

**Published:** 2025-06-28

**Authors:** Mustapha Mouzai, Mohamed Amine Riahla, Amor Keziou, Hacène Fouchal

**Affiliations:** 1LIMOSE Laboratory, University M’Hamed Bougara of Boumerdes, Boumerdes 35000, Algeria; ma.riahla@univ-boumerdes.dz; 2CNRS, LMR, Université de Reims Champagne-Ardenne, 51687 Reims, France; amor.keziou@univ-reims.fr; 3LAB-I*, Université de Reims Champagne-Ardenne, 51687 Reims, France; hacene.fouchal@univ-reims.fr

**Keywords:** GPS spoofing attack, unmanned aerial vehicles, machine learning

## Abstract

All current transportation systems (vehicles, trucks, planes, etc.) rely on the Global Positioning System (GPS) as their main navigation technology. GPS receivers collect signals from multiple satellites and are able to provide more or less accurate positioning. For civilian applications, GPS signals are sent without any encryption system. For this reason, they are vulnerable to various attacks, and the most prevalent one is known as GPS spoofing. The main consequence is the loss of position monitoring, which may increase damage risks in terms of crashes or hijacking. In this study, we focus on UAV (unmanned aerial vehicle) positioning attacks. We first review numerous techniques for detecting and mitigating GPS spoofing attacks, finding that various types of attacks may occur. In the literature, many studies have focused on only one type of attack. We believe that targeting the study of many attacks is crucial for developing efficient mitigation mechanisms. Thus, we have explored a well-known datasetcontaining authentic UAV signals along with spoofed signals (with three types of attacked signals). As a main contribution, we propose a more interpretable approach to exploit the dataset by extracting individual mission sequences, handling non-stationary features, and converting the GPS raw data into a simplified structured format. Then, we design tree-based machine learning algorithms, namely decision tree (DT), random forest (RF), and extreme gradient boosting (XGBoost), for the purpose of classifying signal types and to recognize spoofing attacks. Our main findings are as follows: (a) random forest has significant capability in detecting and classifying GPS spoofing attacks, outperforming the other models. (b) We have been able to detect most types of attacks and distinguish them.

## 1. Introduction

In recent years, unmanned aerial vehicles (UAVs), commonly known as drones, have witnessed a significant surge in use. Driven by their proven efficiency and precision in accomplishing border surveillance and rescue missions, they have been adopted for civilian applications in numerous fields, including agriculture, delivery services, wildlife tracking, the film industry, and beyond [1]. Currently, the size of the unmanned aerial vehicle market is estimated at USD 12.39 billion in 2025, and it is expected to reach USD 20.72 billion by 2030 [2]. These devices are characterized by a diverse set of properties depending on their specific applications, including size, velocity, mobility models, computational capabilities, and battery capacity. Meanwhile, some models are equipped with complementary components, such as an inertial measurement unit (IMU), visual odometry, and optical flow sensors, to expand their functions.

The growing reliance on UAVs is accompanied by significant security and privacy concerns. These systems are highly vulnerable to a wide range of cyberattacks that threaten their integrity, confidentiality, and availability. A systematic classification of cyberattacks against UAVs was presented at four different levels [3]. At the software level, the threats include malicious software and operating system attacks. The hardware level is targeted by hijacking attempts and physical collisions. The communication level faces risks such as denial-of-service (DoS) attacks and node misbehavior. Finally, at the sensor level, UAVs are susceptible to false sensor data injection and GPS jamming. These attacks can severely compromise UAV operations, leading to mission failures, data breaches, and even physical damage.

Our study focuses on sensor-based vulnerabilities, specifically those threatening UAVs’ navigational system, which may be leveraged into hardware-level threats. The complexity of managing UAV flights in dynamic environments, such as in cyberattack situations, requires control systems to continuously interact with multiple sensors over time for the purpose of detecting and isolating raised disturbances. This introduces nonlinearity challenges in modeling motion behavior based on control inputs. Therefore, essential security actions involve learning-based and state-estimation-based methods. In [4], the authors designed a neural network model empowered with an extended Kalman filter for the online detection and isolation of a cyberattack that affects UAVs’ navigation system, specifically involving the injection of false data (FDI) into the inertial measurement unit (IMU) sensor. Ref. [5] introduced a real-time health monitoring framework to detect potential anomalies in the aircraft engine system. Their approach consists initially of applying kernel principal component analysis (KPCA) to extract highly correlated flight variables. This step is critical in reducing computational overhead and preventing overfitting. Then, they trained a support vector regression (SVR) model that predicts flight behavior, which was compared to healthy flights using statistical bounds to identify abnormalities. The work of [6] addressed the challenges of ensuring resilient formation control in the context of drones, where communication may be lost or compromised in adversarial environments. They proposed an approach that integrates Graph Attention Networks (GANs) with deep reinforcement learning (DRL) and leverages neighboring nodes to deceive DoS attacks. The researchers in [7] proposed a state-estimation-based method for the real-time detection and isolation of three main types of cyberattacks, namely random attacks, false data injection attacks, and DoS attacks. Their approach is scalable to any nonlinear aerial system.

In addition, UAVs typically rely on the GPS as their primary navigation system, continuously tracking signals from at least four satellites to accurately determine their precise spatial position at any given time. However, in some cases, these signals are affected by the ionosphere and environmental conditions, producing signal noise or, in more serious scenarios, complete signal loss. In addition, civilian GPS signals remain unsecured and could easily be intercepted and retransmitted. Third-party actors with bad intentions can build receiver–spoofer devices using only commonly available low-cost components to deceive GPS receivers. As a result, this might cause instability in determining the real-time position of GPS assets in general, and disrupt UAVs’ aerial navigation in particular.

Therefore, many cybersecurity researchers showed interest in implementing effective countermeasures to deal with the danger posed by such attacks. The study presented in [8] conducted a systematic review of the literature exploring the existing research dimensions on GPS spoofing attacks in the context of UAVs and flying ad hoc networks. They investigated the existing techniques for conducting different types of GPS spoofing and jamming mechanisms, their impact, and the potential to leverage these attacks as a defense concept, such as preventing civilian UAVs from accessing forbidden areas. Additionally, they explored the existing defense strategies proposed in the literature against these attacks.

Since our interest lies in security precautions and defense countermeasures against GPS spoofing attacks, we proceed in this work to follow the template established in [8] to detect, mitigate, and prevent this threat. The existing countermeasures can be divided into several categories. Onboard device techniques consist of equipping UAVs with additional sensors to assist navigation and help to detect attacks. GPS signal processing techniques investigate anomalies in the characteristics of signals, such as signal strength and angle of arrival. Cryptography methods aim at establishing secure communication links between UAV nodes, satellites, and ground control stations to protect the integrity of signals. Game theory methods also contribute to the detection of GPS spoof attacks by modeling the interactions between targets and attackers as a strategic game, ultimately reaching an equilibrium, such as in the Stackelberg game. On the other hand, the relevance recently demonstrated by machine learning in various fields, such as healthcare [9], agriculture [10], and intelligent transportation systems [11], has attracted considerable attention from cybersecurity researchers [12,13,14]. This is due to its contribution to simplifying tasks, improving decision making, and enhancing overall quality of life, in addition to its ability to achieve high precision and automatically adapt to novel conditions. Therefore, numerous studies showed interest in designing machine learning and deep learning models to face GPS spoofing attacks threatening UAVs. They also provided datasets with different formats, demonstrating the impact of GPS spoofing on aerial navigation systems and helping cybersecurity researchers to combat this threat. The SatUAV dataset [15] includes a large amount of high-resolution aerial photography captured from 13 cities around the world, where 605 realistic scene photos with heterogeneous ground features were taken using a real UAV, while an additional 362 photos were added to the dataset from the senseFly website. Log file GPS spoofing datasets [16,17] comprise high-dimensional sensor readings, primarily from GPS and IMU modules, collected during UAV flights in both benign and spoofed scenarios. They often require the implementation of advanced feature selection techniques due to their complexity. The OpenSky Network dataset [18] contains samples of real-world Air Traffic Surveillance (ATS) data, distributed all over the globe. Another dataset provided by [19] involves the characteristics of GPS signals extracted from an eight-channel GPS receiver mounted on a real UAV while performing real missions. It also contains three types of GPS spoofing attacks simulated using MATLAB.

In this paper, we propose a supervised ML-based detection system to identify GPS spoofing attacks and classify each attack type according to its sophistication level. Tree-based machine learning algorithms are employed, namely a basic decision tree model, a bagging ensemble model known as random forest, and a boosting ensemble model named extreme gradient boosting. The dataset used contains 13 features collected from a multi-channel GPS receiver mounted on an autonomous vehicle through multiple missions. As a preprocessing step, our work involves handling non-stationary features and converting the original dataset dimensions into a more suitable format for our models. We conduct a comparison between the proposed classifiers and the existing work in terms of the accuracy, precision, recall, and F1-score metrics.

The remainder of this work is structured as follows: In Section 2, we discuss previous works that address the problem. In Section 3, we provide the methodology used to design our machine learning classifiers. Section 4 presents the obtained results. Finally, Section 5 concludes the paper.

## 2. Related Works

To design effective defenses against cyberattacks, understanding the attacker’s intent is critical. In the context of GPS spoofing, an attacker may try to force a UAV to collide with other UAVs or obstacles, disregarding the risk of detection. Alternatively, the attacker may aim to capture the UAV while carefully avoiding actions that trigger detection systems. To achieve this, the attacker can either collect GPS signals from one location and re-broadcast them in another or generate GPS signals that mimic authentic satellites. As described in [20], GPS spoofing attacks are distinguished by their strategy and impact on UAV navigation:(a)Simplistic attacks: The basic form of a GPS spoofing attack involves launching GPS signals that carry false position information at higher power than authentic satellite signals. This forces the target GPS receiver to lose lock and perform the re-acquisition process, raising the concern of an ongoing attack. If, by any chance, the attacker successfully prevents the loss of lock, the attack still lacks time synchronization with previous authentic signals. Moreover, it disregards the UAV’s motion, making the spoofed signal appear clearly abnormal to the target receiver. Nevertheless, the basic GPS spoofing attack can still function as a jamming technique, preventing the receiver from acquiring authentic signals.(b)Intermediate attacks: Intermediate-level attacks include using a portable receiver–spoofer device. It is typically built with low-cost commonly available components and runs an open-source GPS signal simulator. An attacker places this device in proximity to the UAV target to estimate its position, velocity, and time. The purpose is to synchronize the spoofed GPS signals with authentic satellite signals, then gradually increase their amplitude, and subtly alter the reported position to divert the target from its intended path. This attack overcomes the limitations of the basic spoofing attack and is capable of deceiving most traditional detection techniques. However, angle-of-arrival discrimination remains the most effective countermeasure against this attack form as the signals seem to originate from the same direction.(c)Sophisticated attacks: In more sophisticated attacks, adversaries broadcast synchronized signals from multiple angles using a series of receiver–spoofer devices that emulate satellite constellations. The complexity of this attack type makes it extremely challenging to detect. Even advanced detection techniques based on angle of arrival are deceived and may incorrectly classify multipath signals as spoofing attacks.

To address this, numerous studies have proposed various techniques to detect and mitigate the risks posed by these attacks. As described in [8], the existing countermeasures can be classified into five main categories:

### 2.1. Onboard Device-Based Techniques

In [21], the authors proposed a method to detect GPS spoofing attacks in both single- and multiple-UAV formations. The proposed scheme involves reporting the self-position of a node N and at least three of its neighboring nodes to a ground control station (GCS) at random intervals. The GCS then verifies the relative positions between node N and its neighbors by comparing their values with the positions obtained from the GPS. Any disparity indicates the presence of a spoofing attack. In the case of a UAV operating in isolation, the system fuses position information provided by the inertial measurement unit (IMU) with GPS data to identify spoofed signals. However, the study assumed the constant presence of the GCS and a stable communication link with UAV nodes, which is challenging to maintain in real-world scenarios.

Ref. [22] proposed a visual odometry-based method to detect sophisticated GPS spoofing attacks. Cameras were used as aiding sensors to capture images of the UAV’s sub-trajectory, which were then compared with the absolute path of the UAV obtained from the GPS. The difference between the two trajectories was calculated using three different dissimilarity measures, namely the Euclidean distance, angular distance, and taxicab distance. This method proved effective against such attacks. However, it had several limitations, particularly in low-light conditions at night or in areas with challenging visual features, such as regions covered by water or snow.

Ref. [23] presented a method to detect GPS spoofing attacks targeting single- or multiple-UAV nodes. In this technique, each UAV in the formation reports its GPS location data to a GCS. If two adjacent nodes report identical locations, this indicates a spoofing attack as spoofed signals typically affect multiple nodes simultaneously. In cases where only a single UAV is targeted, the approach compared the distance between the target UAV and its neighbors obtained from the GPS coordinates with the distance measured using impulse radio ultra-wideband (IR-UWB) or other ranging technologies. Although this method eliminated the need for additional onboard equipment that may drain the UAV’s battery, it still relies on the availability of the GPS and neighboring UAV nodes.

### 2.2. Signal Processing-Based Techniques

In [24], the authors proposed a detection technique based on uplink received signal strength (RSS). UAVs are required to obtain authorization from the Unmanned Aircraft System Traffic Management (UTM) before each flight, after which they must periodically report their telemetry data, including GPS position in a 5G-enabled environment, as mandated by the Federal Aviation Administration (FAA). The UTM then compares the UAV’s reported GPS position with the position calculated using trilateration of distances to at least three base stations determined via RSSI-based distance estimation.

The study proposed in [18] leveraged an existing global network of 700 air traffic control sensors to detect and mitigate GPS spoof attacks. UAVs and aircraft periodically transmit advertisement messages containing their GPS positions to at least four of these air traffic controllers. The system then verifies these positions by comparing them with estimated locations, calculated by multilateration, using the distances between the aircraft and the controllers. These distances are computed using the Time Difference of Arrival (TDoA)-based technique.

A GPS spoofing detection system inspired by burglary scene analysis was proposed in [25] to identify sophisticated GPS spoofing attacks launched from multiple sources. The approach analyzes the absolute signal power and the carrier-to-noise ratio (${C/N}_{0}$) of the received GPS signals to detect anomalies. Since neighboring UAVs also receive propagated spoofing signals, they can act as witnesses to confirm the presence of any attack. In addition, a trust policy is implemented for witness nodes to prevent the dissemination of false information.

### 2.3. Cryptography-Based Techniques

The authors of [26] proposed an architecture based on blockchain technology, where multiple UAVs collaborate to detect GNSS spoofing attacks. Each UAV performs its designated functions, including sharing its location data on the blockchain. These data are later compared with the UAV’s position obtained via cooperative positioning to identify any suspected spoofing attempts. However, this approach requires UAVs to be equipped with a set of radio direction finding (RDF) transmitters and receivers. In addition, the computational overhead introduced by the blockchain infrastructure remains a significant challenge. Ref. [27] proposed a semi-decentralized UAV architecture to prevent GPS spoofing. A leader UAV is elected by the UAV network community through a consensus process for a predefined time period. The leader obtains its GPS position, encrypts it, and broadcasts it to the entire UAV network. The remaining UAVs then decrypt the leader’s position and compute their own positions relative to it.

### 2.4. Game Theory-Based Techniques

Ref. [28] introduced a mathematical framework to detect and mitigate the effects of GPS spoofing attacks on UAVs; the approach models the attacker and a set of UAV strategies as a dynamic Stackelberg game. However, this technique required communication among UAVs to determine the real position of the target UAV through cooperative localization.

A game-theoretic security mechanism was proposed in [29] to detect and mitigate GPS spoofing attacks based on a continuous kernel signaling game. In their model, a GPS receiver and a spoofer were depicted as two players. The receiver player faces a critical decision, either accepting a received GPS signal without verification, potentially exposing itself to deception, or estimating its position based on the signal. On the other hand, the spoofer player strategically injects counterfeit signals in an attempt to manipulate the receiver’s navigation system and mislead its position.

### 2.5. Machine Learning-Based Techniques

In [30], a convolutional neural network (CNN) model was used to detect GPS spoofing attacks. The methodology consisted of comparing the aerial image of the ground taken by a camera mounted on the UAV with its corresponding satellite image retrieved from Google Earth using the GPS coordinates. The CNN model evaluates the similarity between the two images, and any reported mismatch is considered indicative of a spoofing attack. To train and evaluate their model, the authors introduced the SatUAV dataset [15], which comprises paired images of real aerial imagery collected from 13 cities around the world, and corresponding satellite imagery from Google Earth. In addition, the dataset is enriched with real-world aerial images from the senseFly website.

A machine learning algorithm based on XGBoost was proposed in [16]. The model was initially trained offboard using flight logs from IMU and GPS sensors, and its parameters were tuned using the genetic algorithm. After that, it was deployed onboard a quadrotor UAV to adapt it with additional types of sensors and improve the prediction accuracy. Finally, they conducted real-world flight experiments under both hijacked and non-hijacked scenarios to evaluate the effectiveness of their approach.

In [31], the authors compared the performance of five machine learning algorithms in detecting GPS spoofing attacks. They provided a dataset [19] containing thirteen features of real GPS signals collected from different blocks of a GPS receiver and then simulated attack signatures of three types of GPS spoofing with varying levels of sophistication. The evaluation metrics showed that Nu-SVM had the best performance.

Ref. [32] proposed four tree-based machine learning algorithms, namely random forest (RF), gradient boost, XGBoost, and LightGBM, to detect GPS spoofing attacks. The dataset used was the same as that in [31]. The authors used the Spearman correlation for feature selection and differencing to handle non-stationary data since the dataset contained irregular timeseries features. The evaluation metrics showed that XGBoost had the best performance in terms of accuracy, probability of detection, probability of misdetection, and probability of false alarm.

In [33], the authors evaluated various machine learning models using three ensemble learning techniques, namely bagging, stacking, and boosting, to detect GPS spoofing attacks. As a first step, they applied the Min–Max scaler to normalize the data, then used Pearson’s correlation to remove irrelevant features. They also examined features with no static relationship and transformed them into stationary ones. In that study, grid search was applied to find the best hyperparameters, and 10-fold cross-validation was used to assess the proposed models. The authors concluded that the stacking technique yielded the best results.

The study in [34] is a dynamic-based selection system that chooses the best ML model among ten heterogeneous machine learning algorithms to detect the presence or absence of GPS spoofing attacks. In the preprocessing stage, Ref. [34] applied data imputation to remove empty data and categorical data encoding to transform categories of attacks and authentic signals into 1s and 0s, respectively. In that work, a heterogeneous ensemble feature selection technique combined Spearman correlation and information gain to select the relevant features from the dataset. Meanwhile, the Yeo–Johnson transformation was applied to transform the data to fit a Gaussian distribution, and Bayesian optimization was used for hyperparameter tuning. To validate their method, they compared their results with ensemble model techniques, namely bagging, boosting, and stacking, in terms of various metrics including accuracy, probability of detection, probability of misdetection, and probability of false alarm. The authors claimed that their work outperforms the ensemble model techniques.

A comprehensive comparison of three categories of supervised deep learning models, namely deep neural network (DNN), convolutional neural network (CNN), and recurrent neural network (RNN), was presented in [35] to detect GPS spoofing attacks. Therein, mode imputation was applied to replace missing values with the most frequent value, and the Min–Max scaler was used for data normalization. The authors evaluated the models using a 10-time resampling framework and concluded that the U-Net model based on a CNN achieved the best performance.

A stacking ensemble approach comprising machine learning and deep learning algorithms was proposed in [36] against GPS spoofing. In the preprocessing stage, due to the high range of the numerical values of the dataset, the authors used z-score normalization (standardization) to avoid any kind of feature favoritism. Then, they examined the combination of different ML and DL models. Therefore, the SVM–CNN model showed the best performance in terms of accuracy, precision, recall, and F1-score through K-fold cross-validation.

The summary of the state-of-the-art papers is illustrated in Table 1. Driven by the shortcomings outlined above and the advantages offered by machine learning over traditional approaches, we focus on designing an effective ML-based system that operates independently of any additional sensors onboard or external communication with neighboring nodes or ground control stations. Furthermore, we address GPS spoofing detection by classifying each attack type using a large-scale dataset composed of signal samples collected from real-world experiments.

## 3. Methodology

As shown in Figure 1, our proposed system consists of three main phases. First, we extract data sequences from three missions in the dataset, followed by the necessary preprocessing steps, which include handling non-stationary features and converting the dataset into an appropriate shape. In the second phase, we build machine learning-based classifiers, namely decision tree, random forest, and XGBoost, to classify GPS signals into four categories: authentic signals, simple attacks, intermediate attacks, and sophisticated attacks. Finally, we train each classifier on each mission dataset through 50 experiments and evaluate them using various metrics: accuracy, precision, recall, and F1-score. For further assessment, we combine all the mission datasets into a single dataset to evaluate the consistency of each model on large-scale data.

### 3.1. Dataset Selection

When exploring the datasets provided by GPS spoofing detection studies based on machine learning, we observe that the SATUAV dataset introduced in [30] requires additional hardware modifications, most notably a camera mounted on the UAV. However, our approach focuses on standalone ML solutions that operate independently, without relying on any external onboard devices. On the other hand, the log file dataset from [16] demands high computational resources when training the model during UAV missions due to both the high dimensionality of the sensor logs and the overhead of real-time processing. Therefore, the dataset from [31] remains our preferred choice [37] as it provides characteristics of GPS signals collected from real flights. This dataset captures signal anomalies at early stages, even before the signals reach the PVT (position, velocity, time) block of the GPS receiver, enabling earlier detection of spoofing attacks. Furthermore, the dataset includes simulations of three common types of GPS spoofing attacks: simplistic, intermediate, and sophisticated. Identifying these attack types provides critical insight for designing and deploying effective post-detection countermeasures. However, this dataset presents a challenge that consists of handling timeseries features. The primary function of a GPS receiver is to collect satellite signals when available, which causes varying time intervals between the collected signals. Consequently, it is essential to apply appropriate preprocessing techniques to handle timestamp irregularity so that ML models can perform classification correctly.

In the following section, we outline the main stages carried out to construct the GPS spoofing dataset for UAVs:

#### 3.1.1. Materials

Firstly, the authors designed an eight-channel GPS receiver using a Universal Software Radio Peripheral (USRP) unit, a front-end active GPS antenna, and an open-source Global Navigation Satellite System Software-Defined Receiver (GNSS-SDR). This architecture allowed real-time extraction of GPS signal features at critical points of the processing blocks, including acquisition and tracking. Furthermore, the proposed architecture is capable of tracking diverse satellites simultaneously, providing a more efficient analysis of GPS signals and the various types of spoofing attacks.

#### 3.1.2. Feature Extraction

Multiple experiments were performed to simulate both stationary and mobile autonomous vehicles at different sites and altitudes. During each scenario, the researchers extracted thirteen features of GPS signals collected from each channel of the designed GPS receiver; see Table 2. The features were obtained at different levels of signal processing blocks; this includes pre-correlation, during correlation, and post-correlation stages.

#### 3.1.3. Attack Simulation

To advance the procedures of constructing the GPS spoofing dataset, it is crucial to perform real spoofing attacks on the autonomous vehicle. However, conducting such attacks outdoors is illegal and risky. On the other hand, performing indoor attacks may lead to biased results. Therefore, the authors utilized MATLAB to mimic the signature of the previously described spoofing forms on the features of the received GPS signals. The impact of spoofing attacks on signal integrity is described as follows. In simplistic attacks, a spoofer transmits high-power signals, leading to an increased carrier-to-noise ratio (${C/N}_{0}$), as well as introducing unmatched Carrier Phase (Carrie_phase_cycles) and Doppler shift (Carrier_Doppler_hz) values. As a result, the pseudorange measurement (Pseudorang_m) deviates significantly from the previous authentic signals since the spoofer disregards the target’s motion. In intermediate attacks, the spoofer estimates the target’s position and velocity, then transmits spoofed signals with adjusted power levels to make the (${C/N}_{0}$) values resemble those of authentic signals. The spoofer also aligns the code phase and Doppler shift values to maintain a pseudorange measurement showing no anomalies. Consequently, traditional detection techniques relying on signal processing become ineffective. In sophisticated attacks, the spoofer broadcasts multiple jamming and spoofing signals from different angles. The authors simulated distortions in the correlation peaks of multiple channels simultaneously, and introduced a quadrature accumulation shift in the correlator (PQP).

#### 3.1.4. Dataset Format

The dataset is composed of three Excel files. The first file contains thirteen features extracted from raw GPS signals recorded during UAV missions. Each feature column is divided into eight sub-columns representing the values of the specified feature for each channel at the same time. The second file comprises 158,174 GPS signal samples, including simulated attacks. It contains thirteen features along with a target variable (output) that identifies the type of signal. Genuine signals are labeled as 0, while the spoofed are classified as 1, 2, and 3, corresponding to simplistic, intermediate, and sophisticated attacks, respectively. Each feature in this file also follows the eight-channel structure, resulting in a three-dimensional data format. Finally, a simplified 2D version of the dataset is also available, which is a flattened conversion of the original 3D structure. In this version, each row corresponds to a GPS signal from a single channel rather than aggregating signals across all eight channels in one row. However, rows having zero values indicating unlocked channels that are not actively tracking any satellite are excluded from the dataset, resulting in a total of 510,530 GPS signal samples.

### 3.2. Preprocessing

#### 3.2.1. Extract Missions

In our study, we focus on the three-dimensional version of the dataset, which includes thirteen features recorded across eight channels. We closely explore attack simulations as well as the target variable to assess class balance and verify the integrity of the recorded signals. Additionally, we examine the RX feature, which indicates the time in seconds since the GPS receiver has acquired each signal. This analysis allowed us to extract three distinct time ranges corresponding to separate acquisition of GPS signals.

[491,568–492,039.42 s], which covers 471.42 s (7 min, 51 s, and 420 ms).[173,640–174,233.86 s], which covers 593.86 s (9 min, 53 s, and 860 ms).[262,704.02–264,109.68 s], which covers 1405.66 s (23 min, 25 s, and 660 ms).

Finally, for further improvement, we separate these three missions into different files; the details are demonstrated in Table 3.

#### 3.2.2. Differencing Non-Stationary Features

The Time of Week (TOW) feature represents the timestamp, in seconds, assigned by the satellite at the moment the signal message is generated and transmitted. On the other hand, the RX feature represents the time recorded by the GPS receiver when capturing a GPS signal. Both features are correlated and exhibit a continuous upward trend [31], raising a non-stationary phenomenon.

Since the values of mean, median, and variance may not remain constant, extracting meaningful patterns from TOW and RX becomes challenging for most machine learning algorithms that assume data following a stationary distribution. To mitigate this issue, we apply first-order differencing, as shown in Equation (Equation 1), to both features across the eight channels of each mission file:(1)$$\Delta x_{i}=x_{i}-x_{i-1},$$
where $\Delta x_{i}$ represents the first-order difference, $x_{i}$ is a data sample of the non-stationary feature, and $x_{i-1}$ is its preceding value. Furthermore, rows containing unlocked channels (i.e., rows that have only zero values) are removed to ensure data consistency.

Consequently, the features TOW and RX were replaced by their corresponding derivatives DELTA_TOW and DELTA_RX, respectively. Here, DELTA_TOW represents the time step between successive signals transmitted by the satellites, while DELTA_RX describes the time step between successive signals collected by the GPS receiver.

Table 4, Table 5, Table 6 and Table 7 show the different modality occurrences for the new features classified by mission. We can notice that the DELTA_TOW feature has various modalities since the signals are broadcast from different sources. However, DELTA_RX has only two values for each mission. This can be explained by the fact that the signals are received by a single destination.

#### 3.2.3. Dataset Shape Conversion

The original dataset, which includes simulated attacks, is structured in a three-dimensional (3D) format, as illustrated in Figure 2a. In this format, the thirteen features are distributed across the eight channels of the GPS receiver. In our study, we convert the datasets from the three missions from 3D to a two-dimensional (2D) format. This transformation involves converting the channel-based structure into individual samples. Each channel is treated as an independent row. As a result, the number of samples increases, while the number of features (thirteen) remains unchanged, as illustrated in Figure 2b. This approach preserves the chronological order of the RX timestamps and maintains the temporal integrity of the dataset.

## 4. Machine Learning Classifiers

We encountered in this study several challenges related to the dataset, including class imbalance, non-stationary features, varying numerical scales, and the multiclass nature of the problem. Unlike distance-based models such as k-Nearest Neighbor (k-NN), Support Vector Machines (SVMs), and neural networks, which are highly sensitive to feature scales and require standardization or normalization, decision tree-based models are able to natively handle features with varying ranges without rescaling. In addition, normalizing GPS signal data can sometimes disrupt the intrinsic relationships between features. For instance, ${C/N}_{0}$, which reflects signal strength, directly affects the accuracy of pseudorange measurements.

To handle class imbalance, Ref. [35] applied mode imputation by replacing missing values with the most frequent occurrence. However, this method risks over-representing certain values and introducing bias. Therefore, tree-based machine learning algorithms offer built-in mechanisms to highlight informative features through feature importance measures and incorporate class weighting to address class imbalance. This strategy preserves the original feature scales and their relationships while maintaining the integrity of the data.

In our study, to counteract the threat of GPS spoofing attacks, we investigate tree-based models, namely Decision Tree, Random Forest, and XGBoost, implemented using the commands from sklearn.tree import DecisionTreeClassifier, from sklearn.ensemble import RandomForestClassifier, and from xgboost import XGBClassifier, of the Python library Scikit-learn [38].

### 4.1. Decision Tree

Decision trees were initially introduced by [39]. They are non-parametric supervised machine learning algorithms used for both classification and regression tasks. They follow a hierarchical tree-based structure composed of a root node, internal nodes, branches, and leaf nodes. The mechanism of decision trees consists of generating a set of decision rules derived from features to estimate the target variable. Therefore, internal nodes apply criteria such as Gini impurity or entropy to determine the optimal feature split and obtain the best possible classification. However, this approach still suffers from overfitting in some instances.

In this study, we design a decision tree model employing Gini impurity to measure the quality of splits. The maximum tree depth is not specified; in this case, the tree will grow until all the leaves are pure. We set the minimum number of samples required to split an internal node to 2. Additionally, all the available features are considered during the training process to maximize the model’s learning capacity.

### 4.2. Random Forest

Random forest [40] is an ensemble learning algorithm that constructs multiple decision trees, where each tree is trained on a random sample of the data considering random feature selection at each split. The final predictions are made using majority voting and averaging for the classification and regression tasks, respectively. This approach helps to avoid overfitting issues faced in single decision trees. Moreover, random forest provides feature importance scores; this capability is particularly valuable for understanding which GPS signal features contribute most to detect spoofing attacks. Nevertheless, employing a large number of individual decision trees within the ensemble can result in significant computational overhead.

Our random forest model comprises 100 decision trees, with bootstrap sampling to ensure diverse subsets for training each tree. As the previous explained decision tree, Gini impurity is employed, and the maximum depth of trees is not set. However, the square root of the total number of features is applied to reduce correlation between trees.

### 4.3. Extreme Gradient Boosting

Extreme gradient boosting [41] is an advanced ensemble learning algorithm that builds gradient descent decision trees sequentially. For each iteration, it corrects shortcomings from previous models until reaching a final strong learner. In addition, it leverages parallel processing to accelerate computations, making it significantly faster. Furthermore, XGBoost integrates both L1 and L2 regularization to enhance generalization and mitigate overfitting. On the other hand, it may be memory-expensive when dealing with very large datasets or a high number of boosting cycles.

We construct an XGBoost model employing the softmax objective function to address the multiclass classification task. The number of classes is set to four. The model is configured with a default learning rate of 0.3 and 100 trees within the boosting process. In addition, the maximum depth is left unconstrained.

Table 8 summarizes the hyperparameters of the three models used in our experiments. It provides information on the configurations that led to the respective performances.

## 5. Evaluation and Results

In this section, we conduct an in-depth evaluation of our GPS spoofing attack detection system using datasets from the three missions. Initially, we assess the models’ performance on each mission independently using the Monte Carlo cross-validation technique, which involves running 50 randomized train–test splits.

For each case, the dataset is randomly divided into 70% for training and 30% for testing, taking into account variations in dataset size and class distribution. A feature importance analysis is also conducted on each dataset; this step allows identifying the most significant variables and discarding irrelevant features, consequently reducing the training overhead. However, it is important to highlight that stratified sampling is employed during the train–test split, ensuring a more balanced representation of each class in both subsets. Subsequently, all mission records are merged to create a unified dataset, enabling further evaluation of the model’s generalization capability. Finally, we compare the performance of our proposed models before and after feature selection.

The evaluation metrics used to measure the performance of our models are described as follows: (2)$$\begin{array}{cc} \mathrm{Accuracy} & =\frac{\sum_{i=1}^{C}TP_{i}}{\sum_{i=1}^{C}\left(TP_{i}+FP_{i}+FN_{i}\right)}, \end{array}$$(3)$$\begin{array}{cc} \mathrm{Precision} & =\frac{1}{C}\sum_{i=1}^{C}\frac{TP_{i}}{TP_{i}+FP_{i}} \end{array}$$(4)$$\begin{array}{cc} \mathrm{Recall} & =\frac{1}{C}\sum_{i=1}^{C}\frac{TP_{i}}{TP_{i}+FN_{i}}, \end{array}$$(5)$$\begin{array}{cc} F1\text{-}\mathrm{Score} & =\frac{1}{C}\sum_{i=1}^{C}\frac{2\cdot {\mathrm{Precision}}_{i}\cdot {\mathrm{Recall}}_{i}}{{\mathrm{Precision}}_{i}+{\mathrm{Recall}}_{i}}, \end{array}$$
where C is the total number of classes. Accuracy measures the proportion of correct predictions made by the model throughout the dataset. Precision measures the correctness of positive predictions. Recall indicates how well the model finds all positive predictions. F1-score measures the harmonic mean of precision and recall. Since our contribution involves classifying four categories of GPS signals, we use macro-averaging to compute the evaluation metrics. This method treats all classes equally, including minorities, which is suitable for class-imbalanced datasets. Therefore, each metric is calculated independently for all classes, and then the results are averaged. Using these metrics is essential to confirm that the models do not overfit and maintain consistent performance across different training and testing sets in multiclass classification tasks.

### 5.1. Mission 1

The dataset for Mission 1, as illustrated in Figure 3, consists of 60.6% authentic signals and 39.4% attack signals. The attack samples are further classified into 11.5% simplistic attacks, 10.3% intermediate attacks, and 17.6% sophisticated attacks. Figure 4 shows the DELTA_TOW curve over time.

The feature importance parameter provided by our tree-based ML algorithms provides insight into the relevant features that strongly contribute to detecting GPS spoofing attacks regarding this dataset. As demonstrated in Table 9 and Figure 5, the features PD, CP, DO, TCD, PRN, and C/N0 are widely the most important for the three models, having the highest scores. Meanwhile, the remaining features have a low contribution score. Decision tree and random forest rely on PD as the feature with the highest score, followed by CP and then DO, TCD, PRN, and finally C/N0. The rest of the features converge to zero and can be neglected. The XGBoost model, on the other hand, depends first on the PRN feature, followed by PD, DO, CP, TCD, and C/N0 as the strongest features.

For this mission, the decision tree model achieved an average accuracy of 99.943%, a precision of 99.930%, a recall of 99.927%, and an F1-score of 99.929%. On the other hand, the random forest model demonstrated even stronger results, with an average accuracy of 99.979%, a precision of 99.977%, a recall of 99.973%, and an F1-score of 99.975%. Meanwhile, XGBoost recorded the lowest performance among the three models, with an accuracy value of 99.918%, a precision of 99.902%, a recall of 99.895%, and an F1-score of 99.898%. Despite this, the model remained stable and reliable.

### 5.2. Mission 2

The dataset comprises 65.6% authentic signals and 34.4% attack signals, distributed as follows: 12.2% simplistic attacks, 18% intermediate attacks, and 4.2% sophisticated attacks; see Figure 6. The DELTA_TOW values are illustrated in Figure 7.

The feature importance analysis shows that, as in Mission 1, the strongest features in identifying which class a GPS signal is classified into are still PD, CP, DO, TCD, PRN, and C/N0; see Figure 8. The feature importance scores of each model are presented in Table 10. Therefore, the decision tree’s highest-scoring features are ordered as follows: PD remains the most important feature, followed by TCD, PRN, DO, CP, and C/N0. For random forest, the feature order is PD, TCD, DO, CP, C/N0, and PRN. Finally, XGBoost initially relies on PRN, then followed by PD, TCD, DO, CP, and C/N0.

The decision tree model achieved a mean accuracy of 99.965%, precision of 99.933%, recall of 99.953%, and F1-score of 99.943%, with minimal fluctuations in standard deviation. The random forest model outperformed decision tree, achieving an average accuracy of 99.984%, precision of 99.951%, recall of 99.979%, and F1-score of 99.965%. The XGBoost model scored the lowest performance, with an accuracy of 99.960%, precision of 99.919%, recall of 99.945%, and F1-score of 99.932%.

### 5.3. Mission 3

The dataset for Mission 3, illustrated in Figure 9, consists of 84.9 % authentic signals and 15.1% attack signals. The attacks are categorized as 3.5% simplistic, 4.1% intermediate, and 7.5% sophisticated attacks. Figure 10 demonstrates the DELTA_TOW values for this mission.

As shown in Figure 11, the features PD, TCD, PRN, DO, CP, and C/N0 are the most significant among the others for the decision tree model. In random forest, the feature importance parameter displays PD, TCD, DO, CP, C/N0, and PRN with the highest importance scores. Furthermore, XGBoost’s best features are described as follows: PRN, PD, TCD, DO, CP, and C/N0. The scores obtained for each model can be found in Table 11.

The decision tree model achieved an accuracy of 99.990%, precision of 99.978%, recall of 99.969%, and F1-score of 99.974%. The random forest model provided the highest performance, with an accuracy of 99.995%, precision of 99.991%, recall of 99.984%, and F1-score of 99.988%. Meanwhile, XGBoost demonstrated the lowest performance, with 99.894% accuracy, 99.769% precision, 99.764% recall, and 99.766% F1-score.

### 5.4. All Missions Combined

To further assess the generalization of our models, the datasets of all the missions are consolidated into a single one. The resulting distribution comprises 76.8% authentic signals and 23.2% attack signals, with 6.5% simplistic, 7.5% intermediate, and 9.2% sophisticated attacks; see Figure 12.

The investigation of feature importance on the dataset comprising all the missions allows us to observe from Figure 13 that the decision tree model relies on the PD feature as the most relevant, followed by CP, DO, TCD, PRN, and C/N0, while the remaining features converge to zero. The random forest model leverages PD as the main feature, accompanied by TCD, CP, DO PRN, and C/N0; however, the rest of the features could be ignored. For the XGBoost model, it depends mainly on PD, as in the two previous models, followed by PRN, DO, TCD, CP, and C/N0. The discussed scores for each model are presented in Table 12.

Through 50 independent runs of our models on the dataset comprising all the missions, we examine the behavior of each model across varying training and testing sets. Accordingly, we illustrate the results using curves that demonstrate the values of the evaluation metrics in each test. Initially, the analysis of the decision tree model, as illustrated in Figure 14a, shows that the accuracy remains stable, with a small standard deviation (STD) of ±0.006. The worst accuracy observed was 99.951%, the best was 99.988%, and the average was 99.972%. The precision values, shown in Figure 14b, range from a minimum of 99.909% to a maximum of 99.987%, with an average of 99.953% and an STD of ±0.015. For recall, the model achieved a minimum of 99.90% and a maximum of 99.969%, with an average of 99.941% and a low STD of ±0.017, as shown in Figure 14c. Meanwhile, the F1-score, which represents the balance between precision and recall, achieved an average of 99.947%, with a worst value of 99.904%, a best of 99.978%, and a stable performance indicated by an STD of ±0.015, as shown in Figure 14d.

For the random forest model, Figure 15a shows an impressive accuracy, reaching a maximum of 99.996% and a minimum of 99.982%, averaging 99.99% with a very low STD of ±0.002. Figure 15b shows the precision values, ranging from 99.971% to 99.996%, with an average of 99.985% and a low STD of ±0.005. The recall, shown in Figure 15c, reached a maximum of 99.992% and a minimum of 99.965%, with a mean of 99.979% and a stable STD of ±0.006. Furthermore, the F1-score achieved a best value of 99.993%, a worst value of 99.97%, and an average of 99.982%, with an STD of ±0.004, as shown in Figure 15d.

Lastly, the XGBoost model was found to be the least effective. As shown in Figure 16a, it achieved a best accuracy of 99.939% and a worst of 99.842%, with a mean of 99.894% and an STD of ±0.02. Precision, shown in Figure 16b, ranged from 99.731% to 99.886%, with an average of 99.814% and an STD of ±0.04. Figure 16c shows the recall values, with a maximum of 99.915%, a minimum of 99.742%, a mean of 99.828%, and an STD of ±0.03. Finally, the F1-score, illustrated in Figure 16d, had a mean of 99.821%, reaching a maximum of 99.898% and a minimum of 99.736%, with an STD of ±0.03.

As a consequence, all the models achieved high performance and demonstrated stable behavior. However, the random forest model delivered the highest overall results compared to the other models.

### 5.5. Feature Selection

In this section, we apply feature selection on the dataset comprising all the missions in order to extract the relevant features that reduce model complexity while maintaining the performance. Therefore, for each model, we set a threshold of 0.05 as the feature relevancy score and created Table 13 to describe the status of each feature with respect to this threshold. Consequently, the features PRN, DO, PD, CP, and TCD exceed the threshold score and will therefore be preserved. Additionally, we observe that C/N0 is considered important for both the random forest and XGBoost models but not decision tree. Thus, we retained it as a relevant feature.

We ran 50 independent tests of our ML models on the all-missions dataset after the application of feature selection. The results obtained were subsequently summarized. The decision tree model acquired an average accuracy of 99.976%, a precision of 99.962%, a recall value of 99.947%, and F1-score of 99.954%. Random forest achieved 99.99% accuracy, 99.984% precision, 99.98% recall, and 99.982% F1-score. Finally, XGBoost obtained an average accuracy of 99.894% and precision of 99.817%; for the recall value, it obtained 99.828%, and F1-score of 99.822%.

Figure 17, Figure 18 and Figure 19 demonstrate the evaluation metric behavior of our proposed decision tree, random forest, and XGBoost models, respectively. This information was captured at each iteration of the 50 independent tests before and after applying feature selection on the combined missions dataset. As a consequence, this experiment proves that the performance of the three models remains consistent even after applying feature selection.

For additional investigation, we compare the time required for each model to fit the training data before and after feature selection using a system with Python 3.9 running on an Intel Core i5-8365U CPU with 16 GB RAM. Therefore, It is clear from Table 14 that the training time is reduced after discarding irrelevant features. However, among the models, random forest exhibits the longest training time, whereas decision tree demonstrates the fastest convergence.

To summarize, these experimental results prove the efficiency of our models despite the presence of class imbalance and various dataset sizes. The random forest ensemble model based on bagging emerged as the best-performing approach, consistently achieving the best scores across all the dataset parameters, succeeded by the basic decision tree model, while the XGBoost ensemble method achieved the worst results. The results breakdown is presented in Table 15. However, the random forest model showed the longest convergence time, in contrast to decision tree, which was faster. The feature importance parameter provided by tree-based ML models allowed the extraction of the most significant features in detecting GPS spoofing attacks, namely PRN, DO, PD, CP, TCD, and C/N0, consequently reducing the processing time.

Furthermore, our approach, which explores a GPS characteristic dataset to perform multiclass classification of GPS signals using tree-based models, outperforms the state-of-the-art methods based on binary classification [36] and those leveraging satellite imagery datasets [30] for detecting GPS spoofing attacks on UAVs. As shown in Table 16, X refers to studies that do not consider multi-attack classification, while ✓ performs multi-attack classification.

### 5.6. Open Challenges

The promising findings presented by our approach still have significant limitations since the utilized dataset comprises simulated spoofing attacks. Such environments do not reflect the full complexity and uncertainty of real-world spoofing, which may not capture the impact of such a threat. Cyber-adversaries may employ adaptive, stealthy, or coordinated strategies that are difficult to reproduce in structured simulations. Furthermore, there exist external factors that may affect the performance of our ML models, which are not expressed in the dataset; these include urban obstructions, atmospheric disturbances, and electromagnetic interference. Therefore, generalizing our proposed approach to production UAV systems may be impractical. Consequently, it is essential to explore real-world datasets with more sophisticated and dynamic spoofing methods to validate UAVs’ resistance against GPS spoofing attacks.

Another critical issue relates to post-detection strategies. The current studies based on machine learning lack techniques enabling UAVs to predict their position and maintain navigation autonomy when reliable GPS signals are absent in order to prevent mission failure. Furthermore, the existing models are often trained on static datasets and lack adaptability to evolving spoofing tactics and dynamic environmental conditions. Developing adaptive and self-learning models that can continuously be updated based on real-time environmental feedback is crucial for improving the resilience of UAV navigation systems.

Consequently, it is imperative that researchers tackle these challenges in the future to provide resilient and operational GPS spoofing defense systems capable of ensuring the safety of UAVs in adversarial environments.

## 6. Conclusions

In this study, we tackled one of the most prevalent failures related to UAVs’ aerial navigation system, known as a GPS spoofing attack. Indeed, malicious GPS signals may disrupt UAV navigation systems, causing damage. In our study, we worked on a well-known dataset in this area. We extracted data sequences from three missions with different sizes. We handled thirteen features collected from eight channels of a GPS receiver. Then, we evaluated the efficiency of tree-based classifiers, namely decision tree, random forest, and XGBoost, over numerous independent tests in classifying authentic GPS signals against simplistic, intermediate, and sophisticated spoofed signals. Additionally, we leveraged the feature importance attribute of our tree-based models to extract the relevant features in detecting such attacks. Very interesting results were demonstrated by the random forest model, achieving an average accuracy of 99.99%, outperforming the existing methods in the literature. In addition to that, we were able to distinguish precisely between the attack types compared. This finding is very different from those in the literature. Indeed, none of the studied works were able to detect the attack type. However, despite the contribution provided in this work, many challenges require further investigation. In our future work, we intend to assess the generalization capability of our proposed models tuned on simulated GPS spoofing data, with real-world GPS spoofing attacks. We also aim to investigate mitigation spoofing mechanisms in order to help UAVs manage the security of their navigation.

## Figures and Tables

**Figure 1 sensors-25-04045-f001:**
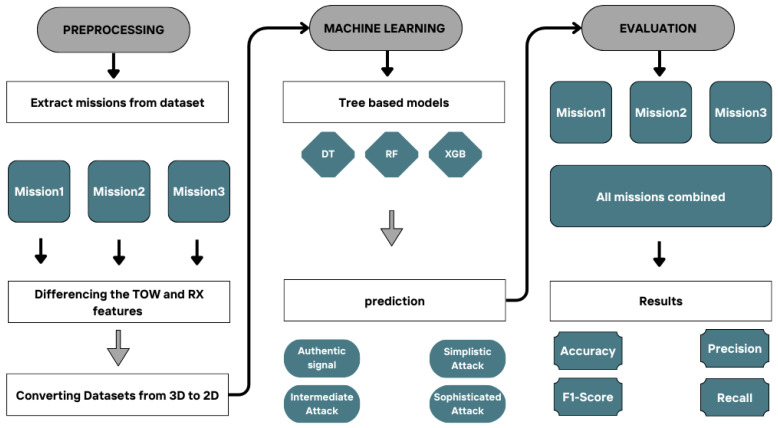
Workflow of our methodology.

**Figure 2 sensors-25-04045-f002:**
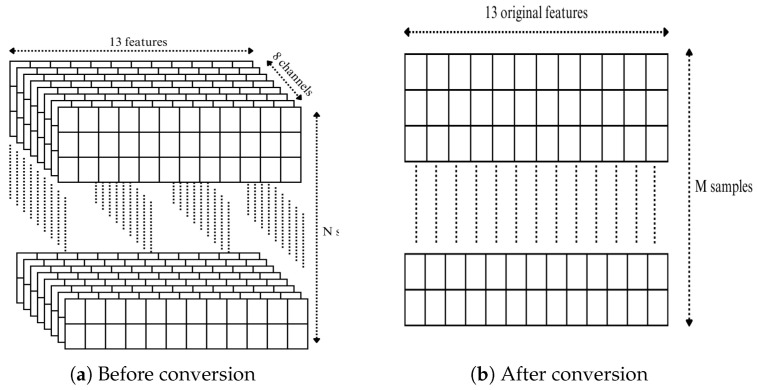
Dataset shapes before and after conversion.

**Figure 3 sensors-25-04045-f003:**
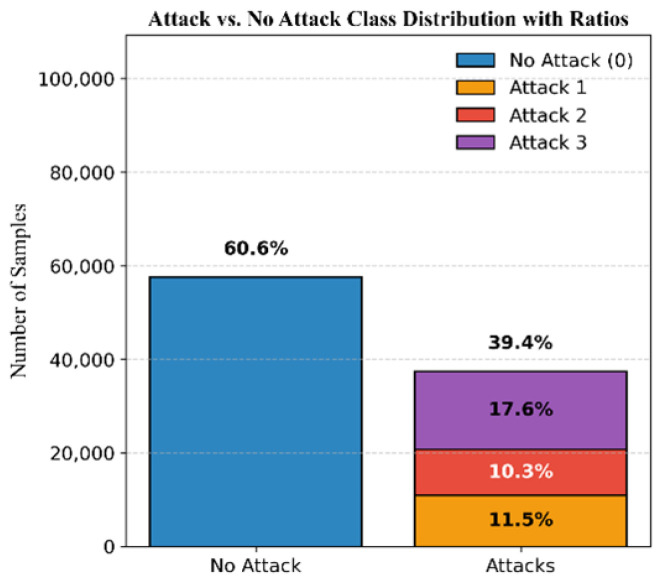
Mission 1 class distribution.

**Figure 4 sensors-25-04045-f004:**
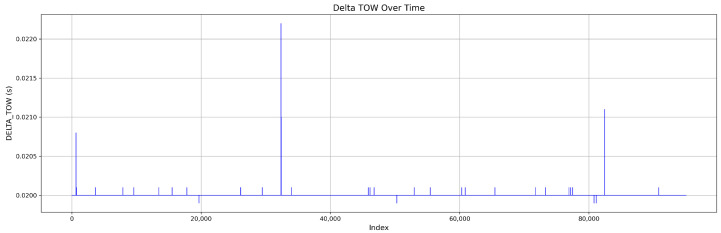
Continuous DELTA_TOW values in Mission 1.

**Figure 5 sensors-25-04045-f005:**
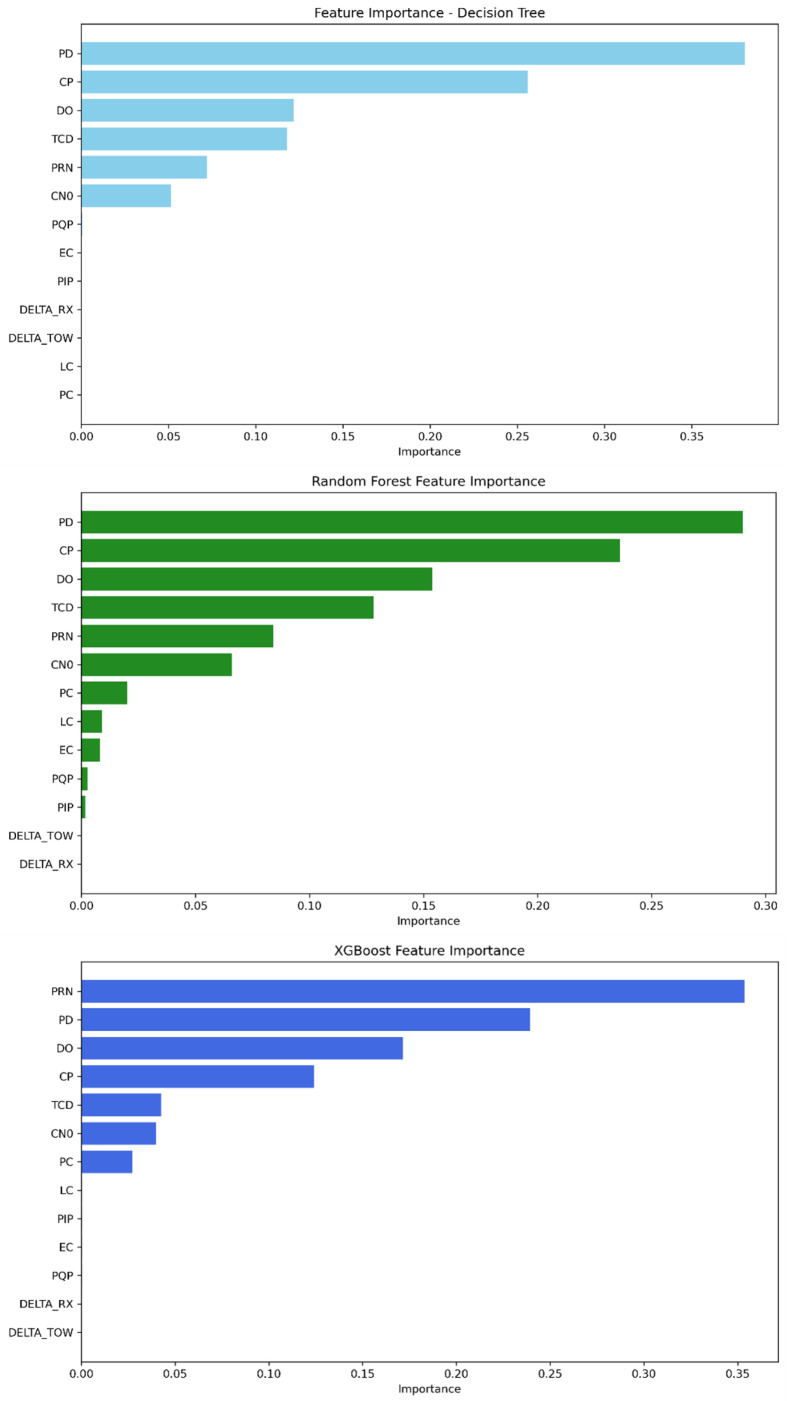
Feature importance ranking from most to least significant—Mission 1.

**Figure 6 sensors-25-04045-f006:**
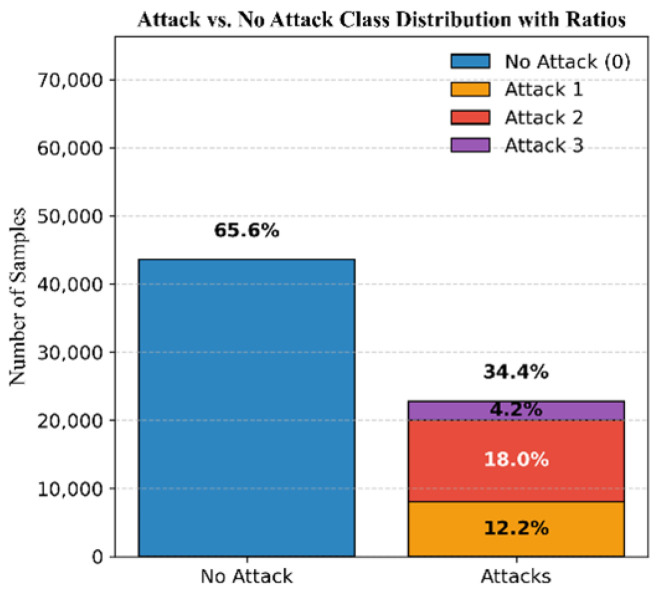
Mission 2 class distribution.

**Figure 7 sensors-25-04045-f007:**
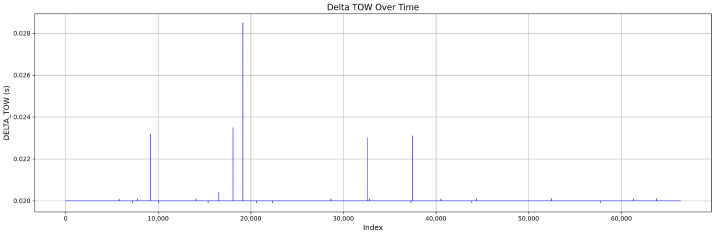
Continuous DELTA_TOW values in Mission 2.

**Figure 8 sensors-25-04045-f008:**
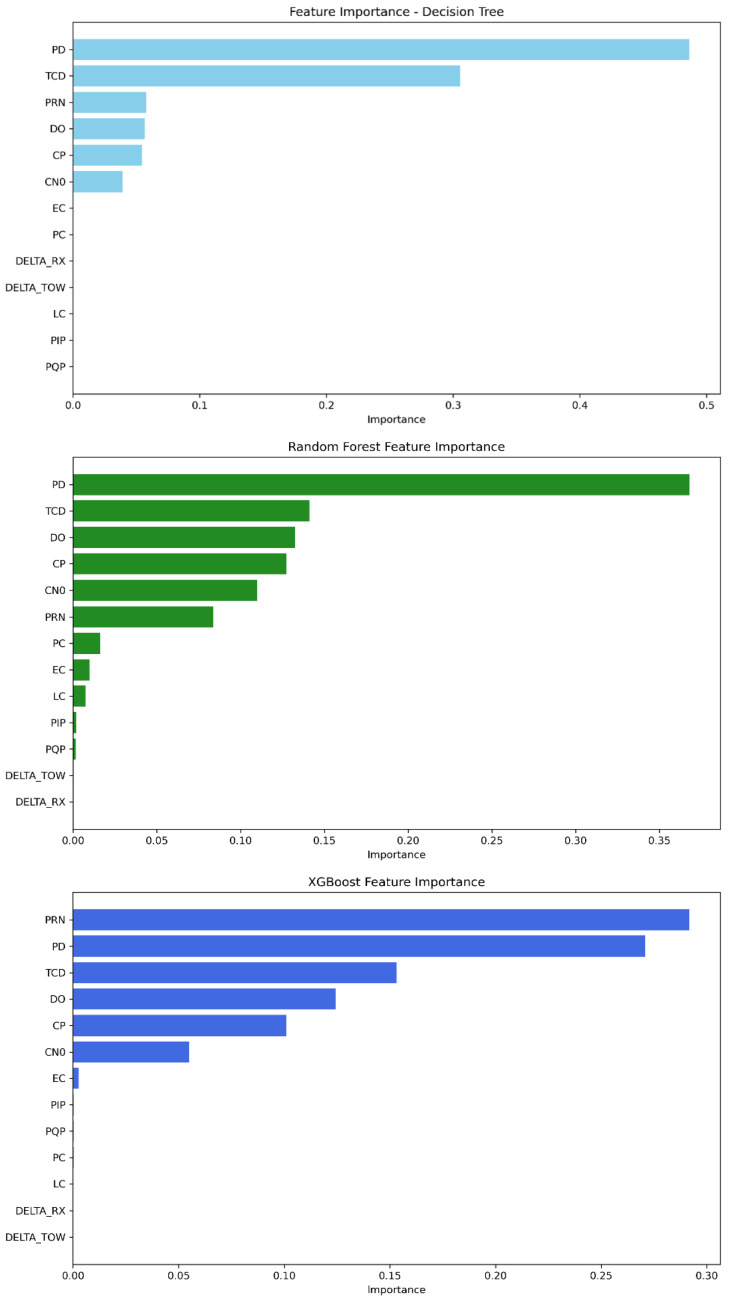
Feature importance ranking from most to least significant—Mission 2.

**Figure 9 sensors-25-04045-f009:**
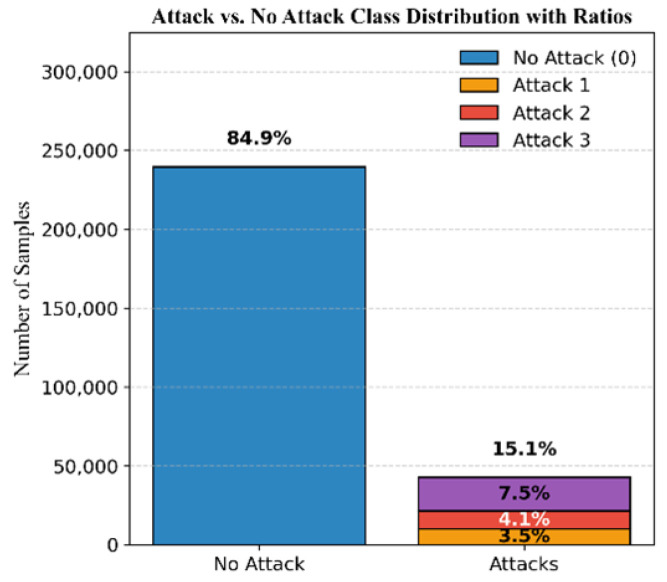
Mission 3 class distribution.

**Figure 10 sensors-25-04045-f010:**
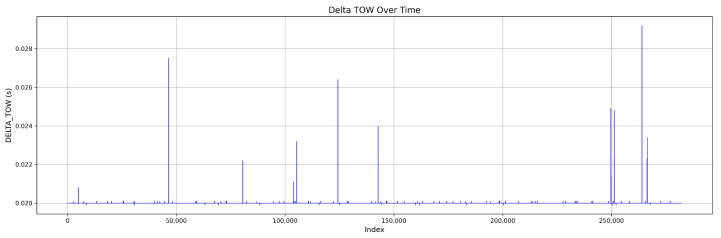
Continuous DELTA_TOW values in Mission 3.

**Figure 11 sensors-25-04045-f011:**
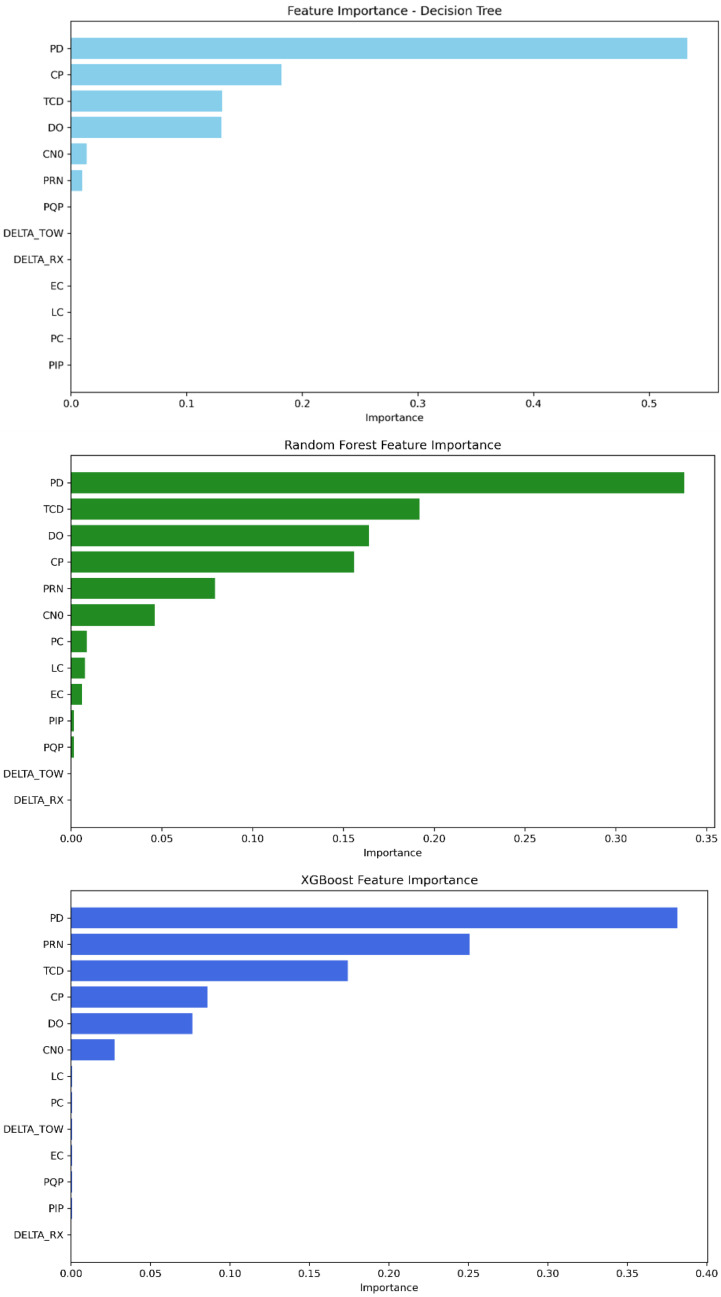
Feature importance ranking from most to least significant—Mission 3.

**Figure 12 sensors-25-04045-f012:**
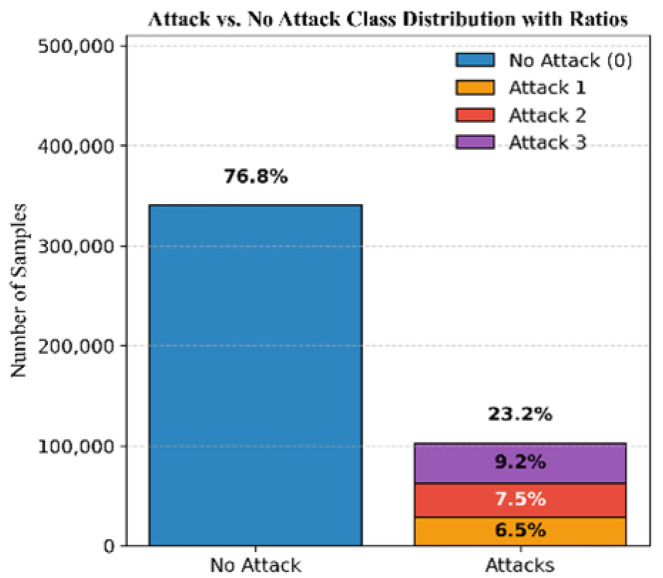
Class distribution of the three missions combined.

**Figure 13 sensors-25-04045-f013:**
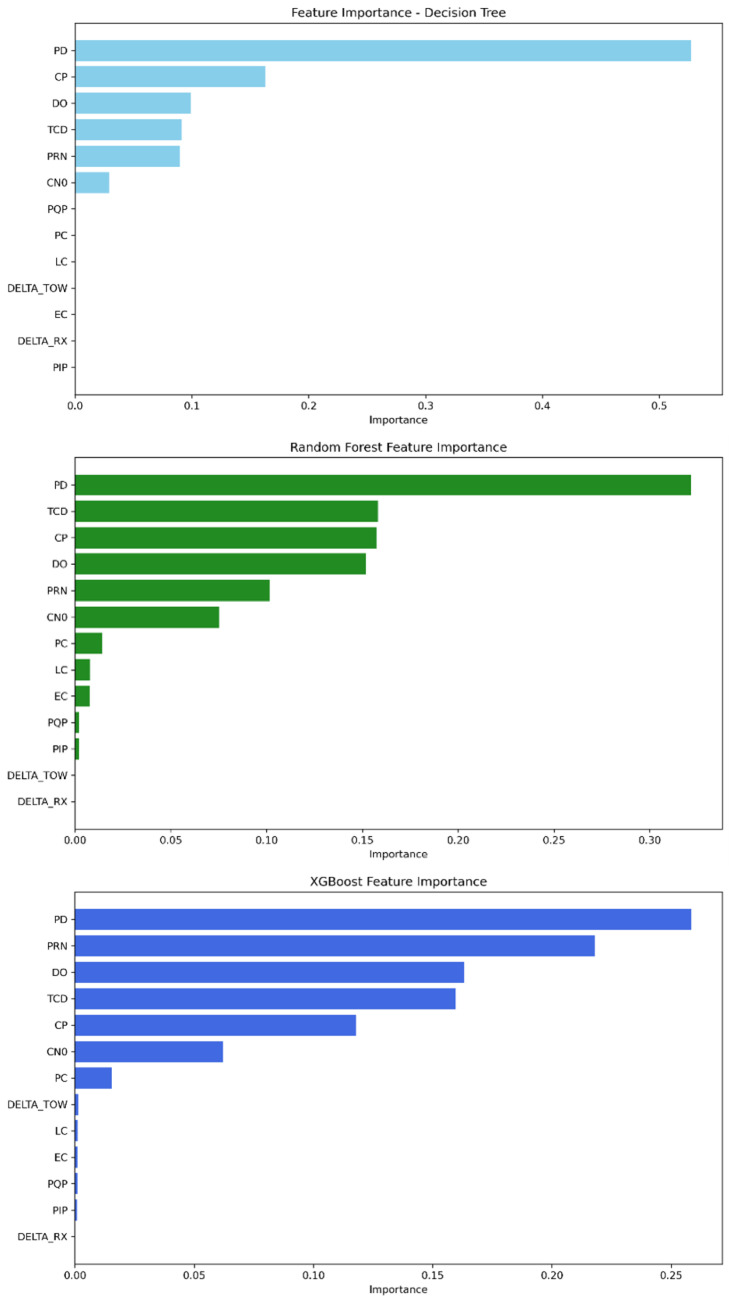
Feature importance ranking from most to least significant—all missions combined.

**Figure 14 sensors-25-04045-f014:**
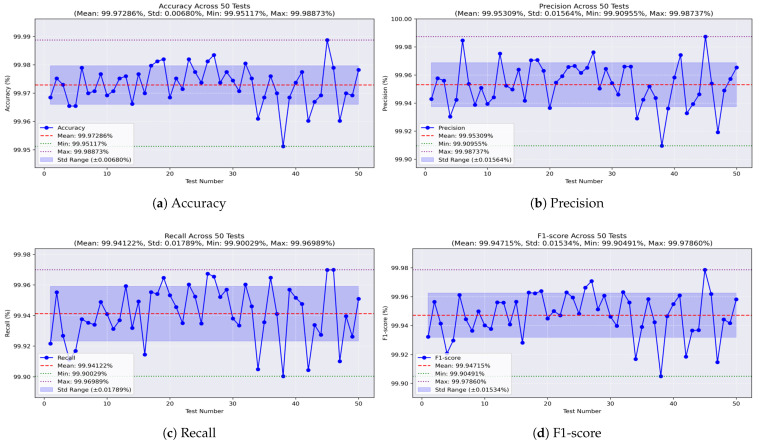
Performance metrics of the decision tree model on all-missions dataset.

**Figure 15 sensors-25-04045-f015:**
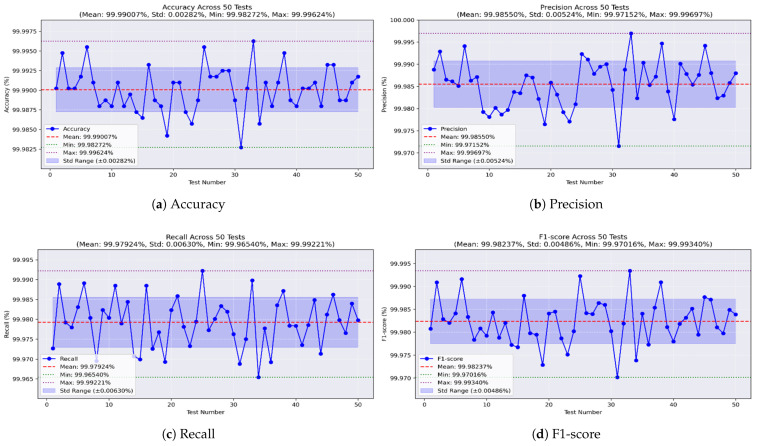
Performance metrics of the random forest model on all-missions dataset.

**Figure 16 sensors-25-04045-f016:**
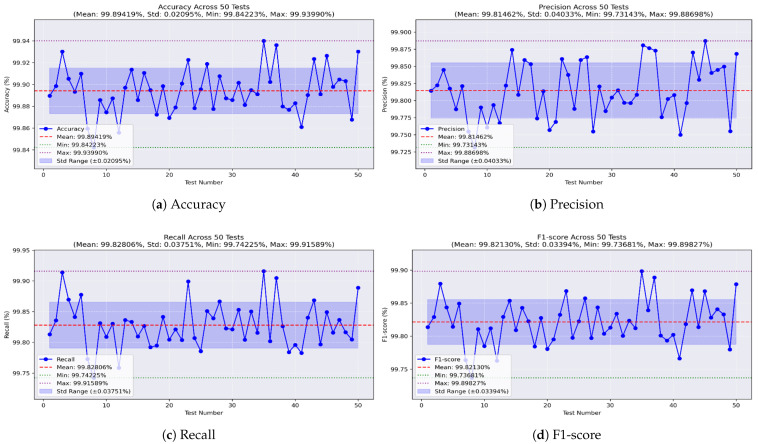
Performance metrics of the XGBoost model on all-missions dataset.

**Figure 17 sensors-25-04045-f017:**
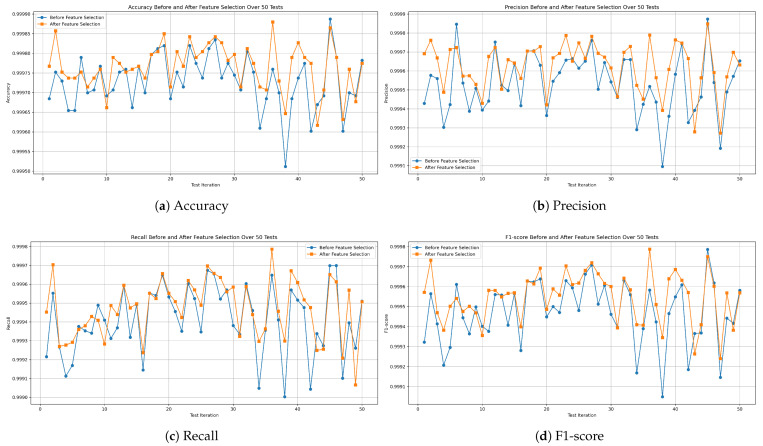
Evaluation metric comparison before and after applying feature selection of the decision tree model.

**Figure 18 sensors-25-04045-f018:**
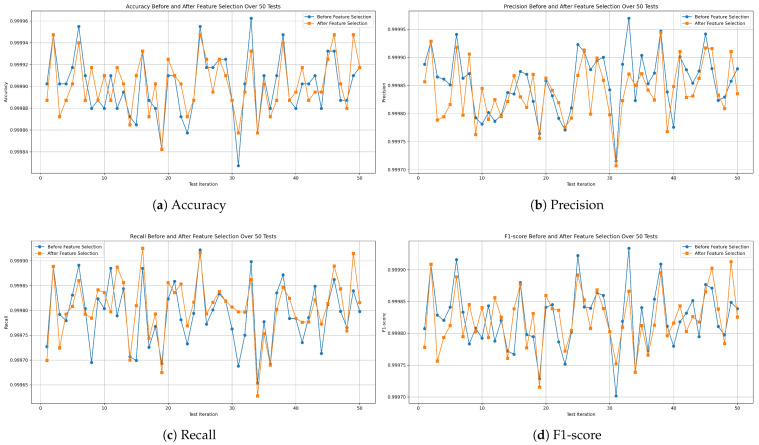
Evaluation metric comparison before and after applying feature selection of the random forest model.

**Figure 19 sensors-25-04045-f019:**
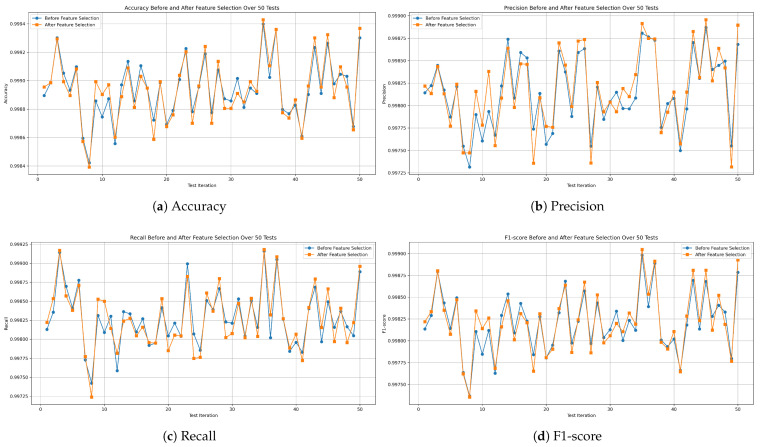
Evaluation metric comparison before and after applying feature selection of the XGBoost model.

**Table 1 sensors-25-04045-t001:** State of the art summary.

Axis	Paper	Methodology	Limitation
Onboard devices	[21]	IMU	
	[22]	Visual odometry	Hardware modification
	[23]	Impulse radio ultra-wideband	
Signal processing	[24]	RSSI	Requires stable link with GCS and neighbor nodes, lack of trust policy among UAV nodes.
	[18]	TDoA	
	[25]	${C/N}_{0}$	
Cryptography	[26]	Blockchain	Computational overhead, complex key distribution in dynamic topologies.
	[27]	Semi-decentralized architecture	
Game theory	[28]	Stackelberg game	Lacks real-time experiments.
	[29]	Signaling game	
Machine learning	[30]	Convolutional neural network (CNN)	Requires camera onboard.
	[16]	Extreme gradient boosting (XGBoost) and genetic algorithm	High-dimensionality dataset, model training overhead.
	[31]	nu-SVM	Lacks classification of attack type.
	[32]	Extreme gradient boosting (XGBoost)	
	[33]	Stacking ensemble approach: KNN, NB, DT, RF, LR	
	[34]	MOD and WMOD ensemble approach: SVM, NB, DT, KNN, LDA, RF, ANN, LR, EN, AdaBoost	
	[35]	CNN: U-net	
	[36]	Stacking ensemble approach: SVM–CNN	

**Table 2 sensors-25-04045-t002:** Features extracted from each channel of the GPS receiver.

Feature Name	Signification	Correlation Level	Description
${C/N}_{0}$	Carrier-to-noise ratio	Pre-correlation	Ratio of the signal power to the noise in dB.
TCD	Tracking Carrier Doppler	During correlation	Doppler shift estimated at the tracking loops in Hz.
PQP	Prompt Quadrature Component		Quadrature component of the prompt correlator (PC).
PIP	Prompt In-Phase Correlator		In-phase component of the prompt correlator (PC).
PC	Prompt Correlator		Correlation measurement between the received satellite signal and the replica signal locally generated.
EC	Early Correlator		1⁄2 chip spacing before the prompt correlator.
LC	Late Correlator		1⁄2 chip spacing after the prompt correlator.
Carrier_phase_cycles	Carrier Phase Cycles	Post-correlation	Beat frequency difference between the received carrier and the replica signal locally generated.
TOW_at_current _symbol_s	Time of the Week in seconds		The number of seconds elapsed given by the satellite atomic clock since the start of each week.
RX_time	Receiver Time		The receiver time given in seconds.
Pseudorange_m	Pseudorange in meters		The distance measured between the receiver and the satellite in meters.
Carrier_Doppler_hz	Carrier Doppler in Hz		Frequency drift between the sent frequency and received frequency due to Doppler effect.
PRN	Pseudorandom Noise		A unique identifier of each satellite.

**Table 3 sensors-25-04045-t003:** Details of the extracted mission files.

Mission	Ranging Time	Period	Size	Class Distribution			
				Label 0	Label 1	Label 2	Label 3
1	[491,568–492,039.42]	471.42 s	95,027	57,599	10,940	9790	16,698
2	[173,640–174,233.86]	593.86 s	66,393	43,572	8072	11,977	2772
3	[262,704.02–264,109.68]	1405.66 s	282,256	239,553	9971	11,495	21,237

**Table 4 sensors-25-04045-t004:** Distribution of DELTA_TOW for Mission 1.

Modality	Occurrence	Modality	Occurrence	Modality	Occurrence
0.0200000000186264	64593	0.019899999955669	2	0.0210000000079162	1
0.0199999999604187	30401	0.0199000000138767	2	0.0211000000126659	1
0.0201000000233761	17	0.0207999999984167	1		
0.0200999999651685	8	0.0222000000067055	1		

**Table 5 sensors-25-04045-t005:** Distribution of DELTA_TOW for Mission 2.

Modality	Occurrence	Modality	Occurrence	Modality	Occurrence
0.0199999999895226	42473	0.0201000000233761	1	0.0210999999835621	1
0.0200000000186264	23894	0.0231999999959953	1	0.0231000000203493	1
0.0200999999942723	9	0.0204000000085216	1		
0.0199000000138767	4	0.0235000000102445	1		
0.0198999999847728	4	0.0284999999857973	1		

**Table 6 sensors-25-04045-t006:** Distribution of DELTA_TOW for Mission 3.

Modality	Occurrence	Modality	Occurrence	Modality	Occurrence
0.0200000000186264	191856	0.019899999955669	1	0.0212000000174157	1
0.0199999999604187	90286	0.0291999999899417	1	0.0247999999555759	1
0.0201000000233761	42	0.026300000026822	1	0.022400000016205	1
0.0200999999651685	34	0.023700000019744	1	0.0234000000054948	1
0.0199000000138767	14	0.0223000000114552	1	0.0239999999757856	1
0.0207999999984167	2	0.0213000000221654	1	0.0213999999687075	1
0.0201999999699182	1	0.0211000000126659	1	0.0248999999603256	1
0.0274999999674037	1	0.0231999999959953	1	0.0205999999889172	1
0.0270000000018626	1	0.0234999999520368	1		
0.0222000000067055	1	0.0263999999733641	1		

**Table 7 sensors-25-04045-t007:** Distribution of DELTA_RX for the three missions.

Mission	Modality	Occurrence
Mission 1	0.0200000000186264	64613
	0.0199999999604187	30414
Mission 2	0.0199999999895226	42488
	0.0200000000186264	23905
Mission 3	0.0200000000186264	191888
	0.0199999999604187	90368

**Table 8 sensors-25-04045-t008:** Hyperparameter settings of the proposed models.

Model	Hyperparameters
Decision Tree	criterion: ’gini’, min_samples_split: 2, random_state: None
Random Forest	criterion: ’gini’, max_features: ’sqrt’, min_samples_split: 2, n_estimators: 100, random_state: None
XGBoost	objective: ’multi:softmax’, num_class: 4

**Table 9 sensors-25-04045-t009:** Feature importance scores for Mission 1.

Feature	Decision Tree	Random Forest	XGBoost
PRN	0.071903	0.084147	0.353905
DO	0.121739	0.153809	0.171473
PD	0.380501	0.290055	0.239213
DELTA_RX	0.000000	0.000000	0.000000
DELTA_TOW	0.000000	0.000012	0.000000
CP	0.256012	0.236168	0.123958
EC	0.000018	0.008147	0.000457
LC	0.000000	0.008953	0.000554
PC	0.000000	0.020156	0.027237
PIP	0.000017	0.001731	0.000481
PQP	0.000471	0.002654	0.000456
TCD	0.118014	0.128118	0.042452
C/N_0	0.051325	0.066049	0.039815

**Table 10 sensors-25-04045-t010:** Feature importance scores for Mission 2.

Feature	Decision Tree	Random Forest	XGBoost
PRN	0.057747	0.083850	0.291810
DO	0.056479	0.132609	0.124330
PD	0.486443	0.367965	0.270811
DELTA_RX	0.000000	0.000000	0.000000
DELTA_TOW	0.000000	0.000029	0.000000
CP	0.054414	0.127390	0.100905
EC	0.000116	0.009787	0.002693
LC	0.000000	0.007440	0.000228
PC	0.000110	0.016208	0.000288
PIP	0.000000	0.001881	0.000341
PQP	0.000000	0.001683	0.000337
TCD	0.305553	0.141088	0.153162
C/N_0	0.039137	0.110069	0.055094

**Table 11 sensors-25-04045-t011:** Feature importance scores for Mission 3.

Feature	Decision Tree	Random Forest	XGBoost
PRN	0.009924	0.079273	0.250672
DO	0.130057	0.164093	0.076245
PD	0.533118	0.337536	0.381509
DELTA_RX	0.000000	0.000000	0.000000
DELTA_TOW	0.000091	0.000057	0.000702
CP	0.182235	0.155711	0.085856
EC	0.000000	0.006048	0.000690
LC	0.000000	0.007628	0.000829
PC	0.000000	0.008709	0.000758
PIP	0.000000	0.001583	0.000545
PQP	0.000389	0.001531	0.000568
TCD	0.130549	0.191832	0.174175
C/N_0	0.013637	0.045999	0.027451

**Table 12 sensors-25-04045-t012:** Feature importance scores for all-missions dataset.

Feature	Decision Tree	Random Forest	XGBoost
PRN	0.089573	0.101541	0.218030
DO	0.099101	0.151993	0.163281
PD	0.527503	0.321745	0.258517
DELTA_RX	0.000000	0.000000	0.000000
DELTA_TOW	0.000035	0.000027	0.001258
CP	0.162943	0.157375	0.117846
EC	0.000010	0.007641	0.001126
LC	0.000054	0.007857	0.001150
PC	0.000127	0.014292	0.015329
PIP	0.000000	0.002072	0.000928
PQP	0.000176	0.002103	0.001045
TCD	0.091270	0.075285	0.159528
C/N_0	0.029206	0.158070	0.061963

**Table 13 sensors-25-04045-t013:** Feature relevancy.

Feature	Decision Tree	Random Forest	XGBoost
PRN	YES	YES	YES
DO	YES	YES	YES
PD	YES	YES	YES
DELTA_RX	NO	NO	NO
DELTA_TOW	NO	NO	NO
CP	YES	YES	YES
EC	NO	NO	NO
LC	NO	NO	NO
PC	NO	NO	NO
PIP	NO	NO	NO
PQP	NO	NO	NO
TCD	YES	YES	YES
C/N_0	NO	YES	YES

**Table 14 sensors-25-04045-t014:** Training time before and after applying feature selection in seconds.

Feature Selection	Decision Tree	Random Forest	XGBoost
Before	4.2450 s	106.3031 s	6.3194 s
After	2.0497 s	61.8106 s	5.4666 s

**Table 15 sensors-25-04045-t015:** Comparative performance of decision tree, random forest, and XGBoost models on different mission datasets.

Dataset	Model	Accuracy ± STD	Precision ± STD	Recall ± STD	F1-Score ± STD
Mission 1	Decision Tree	99.943 ± 0.018	99.93 ± 0.028	99.927 ± 0.024	99.929 ± 0.021
	Random Forest	**99.979** ± **0.007**	**99.977** ± **0.010**	**99.973** ± **0.011**	**99.975** ± **0.008**
	XGBoost	99.918 ± 0.017	99.902 ± 0.023	99.895 ± 0.027	99.898 ± 0.021
Mission 2	Decision Tree	99.965 ± 0.013	99.933 ± 0.035	99.953 ± 0.027	99.943 ± 0.026
	Random Forest	**99.984** ± **0.009**	**99.951** ± **0.035**	**99.979** ± **0.020**	**99.965** ± **0.019**
	XGBoost	99.96 ± 0.016	99.919 ± 0.038	99.945 ± 0.037	99.932 ± 0.030
Mission 3	Decision Tree	99.990 ± 0.003	99.978 ± 0.011	99.969 ± 0.012	99.974 ± 0.009
	Random Forest	**99.995** ± **0.001**	**99.991** ± **0.006**	**99.984** ± **0.008**	**99.988** ± **0.005**
	XGBoost	99.894 ± 0.019	99.769 ± 0.058	99.764 ± 0.046	99.766 ± 0.040
All missions before Feature Selection	Decision Tree	99.972 ± 0.006	99.953 ± 0.015	99.941 ± 0.017	99.947 ± 0.015
	Random Forest	**99.99** ± **0.002**	**99.985** ± **0.005**	**99.979** ± **0.006**	**99.982** ± **0.004**
	XGBoost	99.894 ± 0.020	99.814 ± 0.040	99.828 ± 0.037	99.821 ± 0.033
All missions after Feature Selection	Decision Tree	99.976 ± 0.005	99.962 ± 0.012	99.947 ± 0.015	99.954 ± 0.012
	Random Forest	**99.99** ± **0.002**	**99.984** ± **0.005**	**99.98** ± **0.006**	**99.982** ± **0.004**
	XGBoost	99.894 ± 0.022	99.817 ± 0.043	99.828 ± 0.040	99.822 ± 0.036

**Table 16 sensors-25-04045-t016:** Comparative analysis between our proposed method and state-of-the-art approaches.

Dataset	Reference	Model	Multi Attack Classification	Accuracy	Precision	Recall	F1-Score
SatUAV	[30]	CNN	X	94.80%	93.00%	97.90%	95.40%
Real-time GPS signals	[36]	SVM–CNN	X	99.72%	99.65%	99.77%	99.72%
	Our best model	RF	✓	99.99%	99.98%	99.98%	99.98%

## Data Availability

The dataset used in this paper is publicly available in the link: https://ieee-dataport.org//documents/dataset-gps-spoofing-detection-autonomous-vehicles.

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
