# Peer review of "Exploring Multi-Channel GPS Receivers for Detecting Spoofing Attacks on UAVs Using Machine Learning"

_sensors, 2025, doi:10.3390/s25134045_

Round 1

Reviewer 1 Report

Comments and Suggestions for Authors

1- In Section 3.1.2 (Features extraction), the authors describe the dataset and how it was generated. The authors mention that they chose this dataset because other datasets required additional hardware or had high computational demands. This suggests that the choice of dataset may have been influenced by practical limitations rather than being the most ideal dataset for the research question.

Suggestion: In the Discussion section, include a more thorough analysis of the limitations of using simulated spoofing attacks. Discuss how the findings might generalize (or not) to real-world scenarios with more complex and dynamic spoofing strategies.
Suggestion: If possible, suggest future research directions that involve testing the proposed system with real-world GPS data and spoofing attacks. This could involve collaboration with researchers who have access to such data or designing experiments to collect it.

Suggestion: While the literature review is comprehensive, consider adding a subsection that specifically discusses existing datasets used in GPS spoofing detection research. This would provide a clearer context for the dataset choice.

2- In Section 3.1.3 (Preprocessing), the paper mentions that they handle non-stationary features by differencing the TOW and RX features, but it doesn't go into detail about why these specific features were chosen or if any other feature engineering was explored.

Suggestion: Expand Section 3.1.3 (Preprocessing) to provide more detail on the feature engineering process. Explain the rationale behind differencing the TOW and RX features and discuss any other feature engineering techniques considered.
Suggestion: Consider including a feature importance analysis (e.g., using Random Forest's feature importance attribute) in the Results section to show which features are most relevant for spoofing detection.

Author Response

Dear reviewers

Thanks for your valuable comments which allow us to enhance the quality of the revised paper.

Please find below our responses to the comments raised by the three reviewers. For each of them, we have used different color (red for the first one, blue for the second one and green for the third one) to highlight our response. This is shown either in the current document or the revised version of the paper.

Reviewer1:

Comment1:

- In Section 3.1.2 (Features extraction), the authors describe the dataset and how it was generated. The authors mention that they chose this dataset because other datasets required additional hardware or had high computational demands. This suggests that the choice of dataset may have been influenced by practical limitations rather than being the most ideal dataset for the research question.

Suggestion: In the Discussion section, include a more thorough analysis of the limitations of using simulated spoofing attacks. Discuss how the findings might generalize (or not) to real-world scenarios with more complex and dynamic spoofing strategies.
Suggestion: If possible, suggest future research directions that involve testing the proposed system with real-world GPS data and spoofing attacks. This could involve collaboration with researchers who have access to such data or design experiments to collect it.

Suggestion: While the literature review is comprehensive, consider adding a subsection that specifically discusses existing datasets used in GPS spoofing detection research. This would provide a clearer context for the dataset choice.

Response1:

Thank you for pointing this out, the following points have been added:

  1. We have included more details in 5 open challenges section describing the limitation of the simulated dataset, and how it may not be practical in real world attacks where other factors exist. Changes are highlighted on pages 34-35, lines: 666-676.
  2. We have extended our Conclusion with some hints about future direction in particular issues about dealing with real-world GPS spoofing attacks. Changes are mentioned on page 35, lines: 703-706.
  3. We have added a review on existing datasets from literature in a subsection in Introduction. Change could be found on page 3, lines: 100-115.

Comment2:

In Section 3.1.3 (Preprocessing), the paper mentions that they handle non-stationary features by differencing the TOW and RX features, but it doesn't go into detail about why these specific features were chosen or if any other feature engineering was explored.

Suggestion: Expand Section 3.1.3 (Preprocessing) to provide more detail on the feature engineering process. Explain the rationale behind differencing the TOW and RX features and discuss any other feature engineering techniques considered.
Suggestion: Consider including a feature importance analysis (e.g., using Random Forest's feature importance attribute) in the Results section to show which features are most relevant for spoofing detection.

Response2:

  1. We have referenced from literature works where RX and TOW are correlated and follow a non stationary distribution, this is due to mean, median and variance values which may not remain constant, therefore ML models struggle to learn from such data. Furthermore, we added tables 4, 5, 6 and 7 describing the different modalities of RX and TOW features after their differencing (we can notice their stable values). Changes could be found on page 12, lines: 412-430; Table 4  and  page 13, Tables 5, 6, 7.

Figures 4, 7 and 10 have been added to 5. Evaluation and results section representing the behavior of TOW feature for each mission. Changes could be found on page 17, Figure 4 ; page 19, Figure 7 and page 22, Figure 10.

  1. We have explored feature importance attribute provided by our tree based models, for each mission and for the dataset comprising all missions. This step has allowed us to extract the principal features contributing in the detection of GPS spoofing attacks. We have discussed for each mission, feature importance analysis of each model and we have also provided a table of feature importance score and figures showing top-down bars to sort the importance of each feature. All these improvements are shown as follows :
  • For mission 1: page 17, lines: 533- 541 ; page 17, Table 9 ; page 18, Figure 5
  • For mission 2: page 19, lines: 553- 559 ; page 20, Table 10; page 21, Figure 8
  • For mission 3: page 22, lines: 571- 575 ; page 23, Table 11; page 24, Figure 11
  • All missions together: page 25, lines: 586- 592 ; page 26, Table 12; page 27, Figure 13
  1. We have extended our work by adding 5 feature selection section where we have ignored features having less importance and preserving the performance of our models. Then we have compared our solutions before and after applying feature selection. For this issue, Table 13 and figures 17,18 and 19 have been added. Changes have been made as follows: pages 30-31, lines: 623-643; Table 13 ; pages 32-33, lines: 654- 659 ; Figures 17,18 and 19.

Reviewer 2 Report

Comments and Suggestions for Authors This work presents supervised learning-based GPS spoofing attack classification using public datasets. The results show potential to improve the attack classification capability. However, the work should be improved, including the background and technique details. 1. The parameters and training process of these three models should be provided. 2. The training curves with multiple runs should be provided to verify the training results. 3. The training data and testing data should be separate to test the generalization capability. 4. It suggested to provide the runtime to show the inference speed. 5. The literatures should be enhanced to better the background, such as the works below: [1] Cyber Attack Detection and Isolation for a Quadrotor UAV With Modified Sliding Innovation Sequences. DOI: 10.1109/TVT.2022.3170725 [2] H. Lee, G. Li, A. Rai and A. Chattopadhyay, "Anomaly detection of aircraft system using kernel-based learning algorithm", Proc. AIAA Scitech Forum, pp. 1224, 2019. [3] A. Abbaspour, K. K. Yen, S. Noei and A. Sargolzaei, "Detection of fault data injection attack on UAV using adaptive neural network", Procedia Comput. Sci., vol. 95, pp. 193-200, 2016. [4] Learning Resilient Formation Control of Drones with Graph Attention Network. DOI: 10.1109/JIOT.2025.3554098

Author Response

Dear reviewers

Thanks for your valuable comments which allow us to enhance the quality of the revised paper.

Please find below our responses to the comments raised by the three reviewers. For each of them, we have used different color (red for the first one, blue for the second one and green for the third one) to highlight our response. This is shown either in the current document or the revised version of the paper.

This work presents supervised learning-based GPS spoofing attack classification using public datasets. The results show potential to improve the attack classification capability. However, the work should be improved, including the background and technique details.

Comment1:

The parameters and training process of these three models should be provided. 

Resposne1:

  1. We have provided more details about the architecture of our models. Changes have been made on page 14 as follows: For decision tree: lines: 468- 472 ; For random forest: lines: 483- 486 ; For XGBoost: lines: 495- 498

Comment2:

The training curves with multiple runs should be provided to verify the training results. 

Response2:

  1. the training curves have been provided and have been discussed in section Evaluation and results for the dataset containing all the missions, please refer to: page 28, lines: 593- 622; page 29, Figures 14, 15; page 30, Figure 16.

Comment3:

The training data and testing data should be separate to test the generalization capability. 

Response3:

For each mission, the data have been split into 70% for training and the remaining 30% for testing, please refer to page 15, lines: 507-508.

Comment4:

It suggested to provide the runtime to show the inference speed. 

Response4:

  1. We have added section 5.5 feature selection, in order to extract the relevant feature from our dataset and compare the performance of our models before and after removing non important features. Therefore, we have introduced our working environment and have compared inference speed between our models. Changes could be found on page 33, lines: 644-659  and Table 14.

Comment5:

The literatures should be enhanced to better the background, such as the works below: [1] Cyber Attack Detection and Isolation for a Quadrotor UAV With Modified Sliding Innovation Sequences. DOI: 10.1109/TVT.2022.3170725 [2] H. Lee, G. Li, A. Rai and A. Chattopadhyay, "Anomaly detection of aircraft system using kernel-based learning algorithm", Proc. AIAA Scitech Forum, pp. 1224, 2019. [3] A. Abbaspour, K. K. Yen, S. Noei and A. Sargolzaei, "Detection of fault data injection attack on UAV using adaptive neural network", Procedia Comput. Sci., vol. 95, pp. 193-200, 2016. [4] Learning Resilient Formation Control of Drones with Graph Attention Network. DOI: 10.1109/JIOT.2025.3554098

Response5:

All these references have been added. Please refer to page 2, lines: 45- 67.

Reviewer 3 Report

Comments and Suggestions for Authors

This paper focuses on the problem of GPS spoofing attacks on unmanned aerial vehicles (uavs) , aiming to detect and classify different types of GPS spoofing attacks through machine learning methods. This paper first introduces the wide application of UAV and its dependence on GPS navigation, and points out the vulnerability caused by the unencrypted GPS signal, especially the threat of GPS spoofing attack to UAV navigation system. The authors review the existing GPS spoofing attack detection and mitigation techniques, and find that most studies are only for a single type of attack, while this paper is based on a dataset containing real UAV signals and three spoofing signals, and it can be used to detect and mitigate GPS spoofing attacks, tree-based machine learning algorithms (decision tree, random forest, and XGBoost) are employed to classify signal types to identify these spoofing attacks. The research results show that random forest performs well in detecting and classifying GPS spoofing attacks, outperforming other models, and is able to detect and distinguish most attacks. In addition, the article highlights the importance of conducting research on multiple types of attacks to develop effective mitigation mechanisms. The paper is well-structured and the writings are fine.

However, there are some problems the authors need to pay attention to.

1. The authors used MATLAB to simulate the attack characteristics, and did not explain the difference between the electromagnetic signal characteristics of the real GPS spoofing attack (such as multipath effect, ionospheric disturbance and other physical layer characteristics) . The lack of real environmental disturbances (such as the urban canyon effect) in the experimental data may lead to doubts about the generalization of the model in real scenarios.

2. Only first-order differences were performed on TOW and RX, the stationarity of the sequence after the difference was not verified, and there may be residual trend components affecting the model performance. 

3. The accuracy of random forest in Mission 3 reached 99.990% , but the generalization ability was not verified by cross-validation or independent test sets, and the high accuracy may be derived from data overfitting or too obvious simulated attack characteristics.

4. The article contains some instances of imprecise or ambiguous phrasing. For example, in describing machine learning algorithms, the sentence "Tree-based machine learning algorithms were introduced, namely, a basic Decision Tree model, a bagging ensemble model known as Random Forest and a boosting ensemble model named Extreme Gradient Boosting" used the term "introduced" inaccurately. This should be revised to "Tree-based machine learning algorithms were employed," as "introduced" implies original implementation or innovation, whereas this context describes algorithmic application rather than novel proposal.

Additionally, the phrase "analyzed deeply this dataset through tree-based machine learning algorithms" in the statement "As a main contribution, we have analyzed deeply this dataset..." is overly vague. To enhance clarity and academic rigor, the authors should specify the analytical methodologies employed, such as feature importance analysis, correlation studies, or interpretability frameworks. This would better demonstrate the depth and methodological structure of their investigation, thereby strengthening the paper's scientific validity.

Author Response

Dear reviewers

Thanks for your valuable comments which allow us to enhance the quality of the revised paper.

Please find below our responses to the comments raised by the three reviewers. For each of them, we have used different color (red for the first one, blue for the second one and green for the third one) to highlight our response. This is shown either in the current document or the revised version of the paper.

This paper focuses on the problem of GPS spoofing attacks on unmanned aerial vehicles (uavs) , aiming to detect and classify different types of GPS spoofing attacks through machine learning methods. This paper first introduces the wide application of UAV and its dependence on GPS navigation, and points out the vulnerability caused by the unencrypted GPS signal, especially the threat of GPS spoofing attack to UAV navigation system. The authors review the existing GPS spoofing attack detection and mitigation techniques, and find that most studies are only for a single type of attack, while this paper is based on a dataset containing real UAV signals and three spoofing signals, and it can be used to detect and mitigate GPS spoofing attacks, tree-based machine learning algorithms (decision tree, random forest, and XGBoost) are employed to classify signal types to identify these spoofing attacks. The research results show that random forest performs well in detecting and classifying GPS spoofing attacks, outperforming other models, and is able to detect and distinguish most attacks. In addition, the article highlights the importance of conducting research on multiple types of attacks to develop effective mitigation mechanisms. The paper is well-structured and the writings are fine. However, there are some problems the authors need to pay attention to.

Comment1:

The authors used MATLAB to simulate the attack characteristics, and did not explain the difference between the electromagnetic signal characteristics of the real GPS spoofing attack (such as multipath effect, ionospheric disturbance and other physical layer characteristics) . The lack of real environmental disturbances (such as the urban canyon effect) in the experimental data may lead to doubts about the generalization of the model in real scenarios.

Response1:

  1. The dataset used in our work is a public available dataset (see reference 37), and not created by our team. However, to clarify this issue, we have changed the title of section 1. Dataset description into 3.1. Dataset selection on page 9, line 321.

Comment2:

Only first-order differences were performed on TOW and RX, the stationarity of the sequence after the difference was not verified, and there may be residual trend components affecting the model performance. 

Response2:

  1. We have added tables 4, 5, 6 and 7 describing all the frequent modalities of RX and TOW features for each mission dataset. We can notice there was no residual trend left after our differencing process. Change could be found on page 12, Table 4 and page 13, Tables 5, 6, 7.
  2. Figures 4, 7 and 10 have been added to Evaluation and results section, representing the curve of TOW feature for each mission. Change could be found on page 17, Figure 4 ; page 19, Figure 7 and page 22, Figure 10.

Comment3:

The accuracy of random forest in Mission 3 reached 99.990% , but the generalization ability was not verified by cross-validation or independent test sets, and the high accuracy may be derived from data overfitting or too obvious simulated attack characteristics.

Response3:

The approach conducted in our study consists of assessing our models through 50 independent test iterations with different train test splits for each execution. This method is known as Monte Carlo cross validation. And we have included its description  in our section 5. Evaluation and Result. Change could be found on pages 15-16, lines 504-511.

Comment4:

The article contains some instances of imprecise or ambiguous phrasing. For example, in describing machine learning algorithms, the sentence "Tree-based machine learning algorithms were introduced, namely, a basic Decision Tree model, a bagging ensemble model known as Random Forest and a boosting ensemble model named Extreme Gradient Boosting" used the term "introduced" inaccurately. This should be revised to "Tree-based machine learning algorithms were employed," as "introduced" implies original implementation or innovation, whereas this context describes algorithmic application rather than novel proposal.

Additionally, the phrase "analyzed deeply this dataset through tree-based machine learning algorithms" in the statement "As a main contribution, we have analyzed deeply this dataset..." is overly vague. To enhance clarity and academic rigor, the authors should specify the analytical methodologies employed, such as feature importance analysis, correlation studies, or interpretability frameworks. This would better demonstrate the depth and methodological structure of their investigation, thereby strengthening the paper's scientific validity.

Response4:

  1. This ambiguity has been solved by rephrasing the related sentences mainly on page 3, line 118 .

  1. The abstract has been adapted in order to respond to the reviewer comment on page 3, lines 13-16.

Round 2

Reviewer 1 Report

Comments and Suggestions for Authors

All issues and comments are addressed

Reviewer 2 Report

Comments and Suggestions for Authors

Thank the author for the improvement. The content and figures could be better organized for presentation.

Reviewer 3 Report

Comments and Suggestions for Authors

I am happy with the revision.